# Decreased clot burden is associated with factor XIII Val34Leu polymorphism and better functional outcomes in acute ischemic stroke patients treated with intravenous thrombolysis

István Szegedi[1], Rita Orbán-Kálmándi[2], Attila Nagy[3], Ferenc Sarkady[2], Nikolett Vasas[4], Máté Sik[4], Levente István Lánczi[4], Ervin Berényi[4], László Oláh[1], Alexandra Crişan[5], László Csiba[1,6], Zsuzsa Bagoly[2,6]*

1 Department of Neurology, Faculty of Medicine, Doctoral School of Neuroscience, University of Debrecen, Debrecen, Hungary, 2 Division of Clinical Laboratory Sciences, Department of Laboratory Medicine, Faculty of Medicine, University of Debrecen, Debrecen, Hungary, 3 Department of Preventive Medicine, Faculty of Public Health, University of Debrecen, Debrecen, Hungary, 4 Department of Radiology, Faculty of Medicine, University of Debrecen, Debrecen, Hungary, 5 Department of Neurology, City Hospital of Odorheiu-Secuiesc, Odorheiu Secuiesc, Romania, 6 ELKH-DE Cerebrovascular and Neurodegenerative Research Group, Debrecen, Hungary

* bagoly@med.unideb.hu

## Abstract

### Background

Intravenous thrombolysis using recombinant tissue plasminogen activator remains the mainstay treatment of acute ischemic stroke (AIS), although endovascular treatment is becoming standard of care in case of large vessel occlusions (LVO). To quantify the thrombus burden in LVO, a semiquantitative CT angiography (CTA) grading system, the clot burden score (CBS) can be used. Here we aimed to study the association between CBS and various hemostasis parameters, and to evaluate which parameters are major determinants of thrombolysis outcome.

### Methods

In this single-centered prospective observational case-control study, 200 anterior circulation AIS patients receiving intravenous thrombolysis treatment without thrombectomy were enrolled: 100 AIS patients with LVO (CBS 0–9) and 100 age- and sex-matched AIS patients without LVO (CBS 10). Fibrinogen, α2-plasmin inhibitor, plasminogen, factor XIII and D-dimer were assessed from blood samples taken before and 24 h after thrombolysis, and FXIII-A Val34Leu was genotyped. CBS was calculated using admission CTA. Short-term outcomes were defined based on the change in NIHSS by day 7, long-term outcomes were assessed according to the modified Rankin scale at 3 months post-event.

### Results

Poor outcomes were significantly more frequent in the CBS 0–9 group. Plasminogen activity on admission was significantly higher in the CBS 0–9 group. In a univariate analysis,

**Data Availability Statement:** All relevant data are within the paper and its Supporting Information files.

**Funding:** The research was founded by grants from the National Research, Development and Innovation Office (NKFI) (2019-2.1.11-TÉT-2019-00065 to I.S., FK128582 to Z.B. and K120042 to L.C.), by the GINOP-2.3.2-15-2016-00048 project co-financed by the European Union and the European Regional Development Fund to L.C., and the Hungarian Academy of Sciences (ELKH-DE Cerebrovascular and Neurodegenerative Research Group) to L.C..

**Competing interests:** The authors have declared that no competing interests exist.

significant protective effect of the Leu34 allele against developing larger clots (CBS 0–9) could be demonstrated (OR:0.519; 95%CI:0.298–0.922, p = 0.0227). Multivariate regression analysis revealed that CBS is an independent predictor of short- and long-term functional outcomes, while such effect of the studied hemostasis parameters could not be demonstrated.

## Conclusions

CBS was found to be a significant independent predictor of thrombolysis outcomes. FXIII-A Leu34 carrier status was associated with smaller thrombus burden, which is consistent with the *in vitro* described whole blood clot mass reducing effects of the allele, but the polymorphism had no effect on thrombolysis outcomes.

## Introduction

Acute ischemic stroke (AIS) is a potentially severe vascular disease that leads to disability or death without proper treatment [1]. The patient's only chance of recovery depends on the successful opening of the occluded vessel. As of today, two evidence-based therapies are approved. The endovascular therapy known as mechanical thrombectomy is used more and more frequently in case of large vessel occlusions worldwide, nevertheless, intravenous thrombolysis using recombinant tissue plasminogen activator (rt-PA) is still the mainstay therapy for AIS [2]. Unfortunately, the 4.5-hour-long therapeutic time window is narrow and thus only a fraction of patients are eligible for this therapy. Moreover, thrombolysis is only effective in approximately 30–40% of patients, while in a smaller subset of cases (6–8%) life-threatening side-effects, e.g., intracerebral hemorrhage occurs. These complications cannot be foreseen at the initiation of therapy and their occurrence remains unexplained [3].

The outcome of thrombolysis can be affected by several factors: higher systolic blood pressure on admission, higher NIHSS score on admission, history of coronary artery disease and/or atrial fibrillation all lead to worse functional outcome [4]. Another important prognostic factor is the presence of large-vessel occlusion (LVO) [5]. Approximately 24–46% of acute ischemic strokes are caused by the occlusion of a large vessel [6]. These patients have more severe neurological symptoms on admission and their prognosis is also worse compared to patients without LVO [6].

In order to quantify the thrombus burden in LVO, a semiquantitative computed tomography angiography (CTA) grading system, the so-called "clot burden score" (CBS) has been developed [7]. The CBS is a 10-point system, evaluating the anterior circulation based on the presence of contrast opacification on CTA. Two points each are subtracted for the absence of contrast opacification in the complete cross-section of any part of the proximal M1 segment, distal M1 segment or supraclinoid ICA, and 1 point each for M2 branches, A1 segment and infraclinoid ICA.

Although it is logical to assume that thrombus size affects the outcome of AIS thrombolysis treatment, little is known about the mechanisms controlling thrombus burden. We can surmize that levels of certain hemostasis factors and genetic polymorphisms are likely to be associated with thrombus size and the prognosis of AIS [8–10]. An important regulator of thrombus burden might be coagulation factor XIII (FXIII), a plasma protransglutaminase, a key regulator of fibrinolysis and clot structure. Its active form (FXIIIa) is responsible for the cross-linking of fibrin chains and the cross-linking of α2-antiplasmin to fibrin, providing

mechanical stability to the fibrin clot and protecting it from premature fibrinolysis [11]. FXIII-A has a common genetic alteration: a G->T in codon 103 of the gene, which leads to a change from valine to leucine in position 34 (FXIII-A Val34Leu) [12]. FXIII-A Leu34 allele influences the rate of FXIII activation by thrombin and as a result it affects clot structure. A protective effect of the polymorphism against myocardial infarction and venous thromboembolism has been described by a number of studies and meta-analyses [13–16]. Most recently, by *in vitro* experiments we have identified that the protective effect of the FXIII-A Leu34 allele might be attributed to its decreasing effect of whole blood clot mass in the presence of the elevated fibrinogen levels [17]. On the other hand, *in vivo* data on potential associations between FXIII-A Val34Leu polymorphism and clot burden is still lacking.

Here we aimed to study the association between CBS and various hemostasis parameters, and to evaluate which parameters are major determinants of the outcome of intravenous thrombolysis.

## Materials and methods

### Patients

In this single-centered prospective observational case-control study, anterior circulation AIS patients eligible for intravenous thrombolysis treatment were enrolled at the Stroke Center of the Department of Neurology, University of Debrecen, Hungary. The 2008 ESO guideline was used for the inclusion and exclusion of patients for rt-PA administration [18]. In this case-control design, each AIS patient with LVO (CBS 0–9) was matched with an age- and sex-matched AIS patient without LVO (CBS 10). Patients were selected from 519 consecutive AIS patients admitted to the stroke center over a period of 59 months. All patients received intravenous thrombolysis within the 4.5 h therapeutic time window using rt-PA according to standard protocols [18]. None of the patients underwent mechanical thrombectomy: thrombectomy was either unavailable or the patient was unsuitable to perform the procedure. The diagnosis of AIS was made based on clinical symptoms, brain imaging using non-contrast computed tomography (NCCT) scan and CT angiography (CTA). A follow-up NCCT was performed for every patient 24 h after the thrombolysis. CBS was calculated using admission CTA imaging data, while Alberta Stroke Program Early CT Score (ASPECTS) was calculated at both NCCT examinations by 4 independent radiologists [19]. Baseline clinical and demographic characteristics were recorded for every patient (age, sex, BMI, previous medications, history of cerebrovascular and cardiovascular diseases, cerebrovascular risk factors). Severity of stroke was determined using NIHSS (National Institutes of Health Stroke Scale) on admission and on day 7 as short-term outcome [20]. In order to identify the etiology of stroke, Trial of ORG 10172 in Acute Stroke Treatment (TOAST) criteria was used [21]. The European Cooperative Acute Stroke Study (ECASS) II criteria was applied to grade haemorrhagic transformation as symptomatic (SICH) or asymptomatic intracranial haemorrhage (SICH) [22]. The following outcomes were investigated: 1/ Short-term functional outcome (day 7): poor short-term outcome was defined as a less than 4 points decrease or any increase of NIHSS by day 7 post-event or death by day 7, while a decrease in NIHSS score by at least 4 points or to 0 was defined as favorable outcome [23, 24]. 2/ Long-term functional outcome (90 days post-event): modified Rankin Scale (mRS) at 90 days post-event was used to assess long-term outcomes [24]. Poor long-term outcome was defined by mRS 3–6 [25].

### Blood sampling and laboratory measurements

Peripheral venous blood samples were collected before the administration of intravenous thrombolysis and 24 hours after the event. Routine laboratory examinations (electrolytes,

glucose, renal and liver function, high-sensitivity C-reactive protein (hsCRP), complete blood count) were performed before thrombolysis using standard methods (Roche Diagnostics, Mannheim, Germany and Sysmex Europe GmbH, Hamburg, Germany). For the examination of hemostasis tests, blood samples were collected in tubes containing 0.109 M sodium citrate (Becton Dickinson, Franklin Lane, NJ). Samples were processed immediately and were centrifuged twice at 1500 g, room temperature for 15 min. Screening tests of coagulation (prothrombin time, activated partial thromboplastin time and thrombin time) were performed from freshly separated plasma samples using routine methods on a BCS coagulometer (Siemens Healthcare Diagnostic Products, Marburg, Germany). Fibrinogen concentration was determined using the Clauss assay. For specific hemostasis measurements, plasma aliquots were labelled with a unique code and stored at −80˚C until further assaying in batches. Standard methods were used for measurement of plasminogen and α2-plasmin inhibitor (α2-PI) activity using a BCS coagulometer (Siemens Healthcare Diagnostic Products, Marburg, Germany). D-dimer levels were measured by an immuno-turbidimetric assay (Innovance D-dimer). Plasma levels of FXIII activity were determined by ammonia release assay using a commercially available reagent kit (Technochrome FXIII, Technochlone, Austria). The isolation of DNA was performed from buffy coats of blood samples by QIAamp DNA Blood Mini Kit (Qiagen, Hilden, Germany). FXIII-A Val34Leu polymorphism (c.103 G > T; rs5985) was determined by real-time PCR using fluorescence resonance energy transfer detection and melting curve analysis on a LightCycler® 480 instrument (Roche Diagnostics GmbH, Mannheim, Germany) [26]. All primers are available from the authors upon request. Investigators performing laboratory measurements were blinded to patient identification and clinical data.

## Statistical analysis

Statistical analysis was performed using the Statistical Package for Social Sciences (SPSS, Release 22.0, Chicago, IL), GraphPad Prism 8.0 (GraphPad Prism Inc., La Jolla, CA) and Stata 12 (Stata Corp, College Station, TX). Shapiro-Wilk test was used to evaluate the normality of the data. For two-group analyses Student's t test (parametric data) or Mann–Whitney U test (non-parametric data) were performed. Kruskal–Wallis analysis with Dunn–Bonferroni post hoc test was used for multiple comparisons. χ2 test or Fisher's exact were used for the evaluation of differences between categorical variables. Binary backward logistic regression model was used to determine the independent predictors of unfavourable short- and long-term outcome after thrombolysis. Results of the logistic regression analysis were expressed as odds ratio (OR) and 95% confidence interval (CI). A P-value of <0.05 was considered statistically significant.

## Informed consent

The study was approved by the Ethics Committee of the University of Debrecen, Hungary and the Ethics Committee of the National Medical Research Council. The study protocol conformed to the ethical guidelines of the 1975 Declaration of Helsinki. All patients or their relatives provided written informed consent.

## Results

In this single-centered prospective observational case-control study, 200 patients with anterior circulation AIS undergoing intravenous thrombolysis by rt-PA were enrolled: 100 patients with LVO (CBS 0–9) and 100 age- and sex-matched patients without LVO (CBS 10) (Table 1). Smoking was significantly more frequent and NIHSS was significantly higher in the CBS 0–9 group. There were significant differences in both radiological and clinical outcomes between

**Table 1. Baseline characteristics of patients according to their CBS.**

| | CBS 0–9 | CBS 10 | p |
|---|---|---|---|
| **Number of patients** | 100 | 100 | |
| **Age (years), median (IQR)** | 71 (62–79) | 69 (62–76) | 0.156 |
| **Male, n (%)** | 51 (51) | 51 (51) | 0.671 |
| **Cerebrovascular risk factors, n (%)** | | | |
| Arterial hypertension | 85 (85) | 86 (86) | 0.841 |
| Atrial fibrillation | 26 (26) | 16 (16) | 0.083 |
| Previous stroke | 20 (20) | 28 (28) | 0.185 |
| Hyperlipidaemia | 56 (56) | 66 (66) | 0.147 |
| Diabetes mellitus | 25 (25) | 28 (28) | 0.631 |
| **Smoking, n (%)** | | | |
| Non-smoker | 59 (59) | 71 (71) | 0.002 |
| Previous smoker | 16 (16) | 2 (2) | |
| Current smoker | 19 (19) | 26 (26) | |
| Undetermined | 6 (6) | 1 (1) | |
| **BMI, mean (SD)** | 28.1 (5.2) | 27.9 (6.5) | 0.766 |
| **Duration of thrombolysis (min), median (IQR)** | 60 (60–62) | 60 (60–60) | 0.234 |
| **Time-to-treatment (min), mean (SD)** | 143.5 (115–176.3) | 145 (109.3–182.8) | 0.523 |
| **rt-PA dose (mg), median (IQR)** | 67 (56.75–81.25) | 68 (58–80) | 0.988 |
| **NIHSS, median (IQR)** | 11 (8–16) | 7 (4–10) | <0.001 |
| **Medication at enrollment, n (%)** | | | |
| Antihypertensive therapy | 76 (76) | 62 (62) | 0.032 |
| Antiplatelet drug | 40 (40) | 37 (37) | 0.663 |
| Anticoagulant drug | 5 (5) | 6 (6) | 0.756 |
| Lipid lowering therapy | 24 (24) | 24 (24) | >0.999 |
| Antidiabetic therapy | 15 (15) | 15 (15) | >0.999 |
| **Laboratory measurements, median (IQR)** | | | |
| INR | 0.96 (0.94–1.04) | 0.97 (0.93–1.03) | 0.613 |
| APTT (sec) | 27.70 (25.90–31.20) | 27.65 (25.68–30.02) | 0.583 |
| WBC (G/L) | 7.84 (6.44–9.40) | 8.02 (6.52–9.54) | 0.530 |
| RBC (T/L), mean (SD) | 4.4 (0.5) | 4.6 (0.6) | 0.035 |
| Hemoglobin (g/L), mean (SD) | 133.6 (16.0) | 139.2 (16.2) | 0.016 |
| Platelets (G/L) | 215.00 (177.00–266.50) | 212.50 (179.00–254.50) | 0.438 |
| Serum glucose (mmol/l) | 6.70 (5.80–8.08) | 6.40 (5.63–7.88) | 0.439 |
| hsCRP (mg/L) | 4.30 (2.14–8.58) | 2.63 (1.37–6.01) | 0.017 |
| Creatinine (umol/L) | 78.00 (66.25–95.75) | 78.50 (68.00–90.00) | 0.870 |
| Plasminogen activity on admission (% | 111.50 (92.00–132.80) | 97.00 (88.00–118.00) | 0.032 |
| Plasminogen activity at 24 h (%) | 91.00 (78.00–105.00) | 89.00 (81.00–101.00) | 0.728 |
| $\alpha$2-plasmin inhibitor activity on admission (%) | 101.00 (94.00–109.80) | 101.00 (88.75–109.00) | 0.494 |
| $\alpha$2-plasmin inhibitor activity at 24 h (%) | 74.00 (67.00–86.00) | 76.00 (67.00–86.00) | 0.305 |
| D-dimer level on admission (mg/L) | 0.82 (0.53–1.46) | 0.70 (0.50–1.60) | 0.431 |
| D-dimer level at 24 h (mg/L) | 2.69 (1.37–4.92) | 1.85 (1.03–5.17) | 0.101 |
| Fibrinogen level on admission (g/L) | 3.88 (3.47–4.62) | 3.91 (3.20–4.55) | 0.468 |
| Fibrinogen at 24 h (g/L) | 3.73 (2.99–4.31) | 3.76 (3.18–4.31) | 0.848 |
| Factor XIII activity on admission (%) mean (SD) | 119.50 (33.77) | 128.70 (51.53) | 0.137 |
| Factor XIII activity at 24 h (%) | 114.10 (86.3–132.5) | 104.30 (80.61–130.3) | 0.168 |
| **Stroke etiology (TOAST), n (%)** | | | |

*(Continued)*

**Table 1.** (Continued)

|  | CBS 0–9 | CBS 10 | p |
|---|---|---|---|
| **Large-artery atherosclerosis** | 72 (72) | 22 (22) | <0.001 |
| **Small-vessel occlusion** | 0 (0) | 18 (18) | |
| **Cardioembolic** | 16 (16) | 16 (16) | |
| **Other/undetermined** | 12 (12) | 44 (44) | |
| **Imaging data, n (%)** | | | |
| **ASPECTS on admission** | | | |
| 0–7 | 4 (4) | 0 (0) | 0.059 |
| 8–10 | 93 (96) | 98 (100) | |
| **ASPECTS at 24h after thrombolysis** | | | |
| 0–7 | 51 (53) | 8 (8) | <0.001 |
| 8–10 | 46 (47) | 90 (92) | |
| **Outcomes, n (%)** | | | |
| **Short-term outcome (ΔNIHSS by day 7)** | | | |
| Good outcome (at least -4 points or NIHSS = 0) | 39 (39) | 53 (53) | 0.001 |
| Unchanged status (± 3 points) | 33 (33) | 35 (35) | |
| Poor outcome (+4 points or more) | 28 (28) | 8 (8) | |
| Undetermined | 0 (0) | 4 (4) | |
| **Long-term outcome (90 days)** | | | |
| mRS 0–2 | 34 (34) | 67 (67) | <0.001 |
| mRS 3–6 | 61 (61) | 31 (31) | |
| Undetermined | 5 (5) | 2 (2) | |
| **Intracranial hemorrhage (ECASS II)** | | | |
| No hemorrhage | 89 (89) | 93 (93) | 0.226 |
| aSICH | 4 (4) | 5 (5) | |
| SICH | 7 (7) | 2 (2) | |
| **Factor XIII-A Leu34 carrier n (%)** | 36 (36) | 52 (52) | 0.023 |

APTT, activated partial thromboplastin time; aSICH, asymptomatic intracerebral hemorrhage; ASPECTS, The Alberta Stroke Program early CT score; CBS, clot burden score; ECASS II, European Co-operative Acute Stroke Study-II; hsCRP high sensitive CRP; INR, international normalized ratio; IQR, interquartile range, mRS, modified Rankin Scale; NIHSS, National Institutes of Health Stroke Scale; RBC, red blood cell count; rt-PA, recombinant tissue plasminogen activator; SD, standard deviation; SICH, symptomatic intracerebral hemorrhage; TOAST, Trial of ORG 10172 in Acute Stroke Treatment; WBC white blood cell count

the two groups. Worse ASPECTS at 24 h post-event and poor short-term and long-term outcomes were significantly more frequent in the CBS 0–9 group. Of the measured routine laboratory parameters, hsCRP was significantly increased in the CBS 0–9 group. Plasminogen activity on admission was significantly higher in the CBS 0–9 group, while other parameters were not significantly different between the two groups. Genotype frequencies of FXIII-A Val34Leu polymorphism were consistent with Hardy-Weinberg equilibrium in the total cohort (FXIII-A Val34Val: n = 112 (56%), FXIII-A Val34Leu: n = 78 (39%) and FXIII-A Leu34Leu: n = 10 (5%)). Allele frequencies of the FXIII-A Val34Leu polymorphism in this cohort did not significantly differ from a large cohort of population control group tested earlier [27]. Factor XIII-A Leu34 allele was significantly more frequent in the CBS 10 group, which is consistent with the *in vitro* described whole blood clot mass reducing effects of the allele. In a univariate analysis, a significant protective effect of the Leu34 allele against developing larger clots (CBS 0–9) could be demonstrated (OR: 0.519; 95%CI: 0.298–0.922, p = 0.0227). Given the modifying effects of fibrinogen concentration on FXIII-A Val34Leu genotype dependent clot structure and thrombus burden, we further looked at the specific relationships

between admission fibrinogen concentration (below or above 3.5 g/L) and the CBS for each FXIII-A genotype, but no significant difference was found between groups (S1 Table). In the CBS 0–9 group, 11 patients (11%) suffered post-lysis ICH, while in the CBS 10 group this complication occurred in 7 patients (7%). Differences between CBS groups were not significant when ICH was considered according to ECASS II (aSICH or SICH), suggesting that clot burden was not associated with post-lysis bleeding complications in the studied cohort of patients.

In order to learn more on the potential associations between clot burden and key factors/ markers of fibrinolysis, the patient group with CBS 0–9 was further divided to subgroups according to the degree of clot burden (S2 Table). None of the investigated hemostasis parameters showed significant associations with the extent of clot burden at admission or at 24 h post-event, which rules out the possibility of excessive consumption in case of larger AIS thrombi. D-dimer levels 24 h post-event did not show significant differences between subgroups of different CBS. Taken together, this suggests that the degree of clot burden is not associated with the investigated fibrinolysis factors/markers and these measurements would not be useful in the clinical practice to estimate thrombus size. To find out which clinical or laboratory parameters are associated with thrombolysis outcomes in the study cohort, a binary backward multiple logistic regression analysis was performed. Univariate analysis revealed that low CBS, higher NIHSS at admission, history of previous stroke, and higher plasminogen at admission were significantly associated with poor short-term outcome (Table 2). Although the FXIII 34Leu allele was significantly more frequent in the patient group with CBS 10, the polymorphism showed no association with short-term outcomes. Based on the logistic regression model including age, sex, previous stroke, NIHSS on admission, plasminogen activity on admission, and CBS, only age, NIHSS on admission and CBS were identified as independent predictors of poor short-term outcome (Table 3).

As for long-term outcomes, univariate analysis revealed that older age, diabetes mellitus, lower CBS, higher NIHSS at admission, elevated serum glucose, elevated CRP, elevated creatinine, higher admission D-dimer value and higher admission fibrinogen levels and lower FXIII activity 24 h post-lysis were significantly associated with poor-long term outcomes (Table 4). Similarly to what was observed in case of short-term outcomes, FXIII-Val34Leu polymorphism showed no association with long-term outcomes in this patient cohort. In a binary backward multiple regression model including all variables that showed significant differences between long-term outcomes, CBS proved to be an independent predictor of long-term functional outcomes (CBS 0–9 vs. 10: OR: 2.501; 95%CI: 1.179–5.306, p = 0.017), among with age, NIHSS on admission and creatinine levels at admission (Table 5).

## Discussion

Understanding the mechanisms driving thrombus burden are critical for improving acute stroke care. The CBS grades thrombus size and length accurately and has been proposed to be useful in identifying poor responders to intravenous thrombolysis treatment [7, 28]. Here we confirm that a higher CBS is associated with better short-term and long-term outcomes and CBS might be used as a prognostic marker of AIS thrombolysis treatment. The results of our study suggest that recanalization and thrombolysis outcome depend greatly on the size of the thrombus. Although a number of studies have proposed CBS to be useful in predicting stroke outcomes, our study is among the first to measure key hemostasis parameters that could be associated with thrombus size in AIS patients. It is plausible to think that thrombus size directly relates to major coagulation or fibrinolysis proteins regulating clot structure and lysis. Interestingly, here we show that with the exception of plasminogen levels at admission, there is no direct relation between thrombus size and the investigated fibrinolysis parameters at

**Table 2. Baseline parameters and specific hemostasis markers according to short-term functional outcomes in the studied cohort.**

| | Favorable outcome | Unfavorable outcome | p |
|---|---|---|---|
| Number of patients | 92 | 104 | |
| Age (years), median (IQR) | 68 (62–76) | 70 (63–79) | 0.108 |
| Male, n (%) | 47 (51) | 57 (55) | 0.602 |
| **Cerebrovascular risk factors, n (%)** | | | |
| Arterial hypertension | 80 (87) | 87 (84) | 0.516 |
| Atrial fibrillation | 17 (18) | 25 (24) | 0.344 |
| Previous stroke | 28 (30) | 19 (18) | 0.047 |
| Hyperlipidaemia | 56 (61) | 64 (62) | 0.924 |
| Diabetes mellitus | 18 (20) | 31 (30) | 0.098 |
| **Smoking, n (%)** | | | |
| Non-smoker | 60 (65) | 66 (63) | 0.092 |
| Previous smoker | 4 (5) | 14 (13) | |
| Current smoker | 23 (25) | 22 (21) | |
| Undetermined | 5 (5) | 2 (2) | |
| BMI, median (IQR) | 27.2 (23.8–31.5) | 27.6 (24.6–30.9) | 0.794 |
| Duration of thrombolysis (min), median (IQR) | 60 (60–60) | 60 (60–60) | 0.603 |
| Time-to-treatment (min), mean (SD) | 139 (104–173) | 145 (115–181) | 0.362 |
| rt-PA dose (mg), median (IQR) | 67 (58–80) | 69.5 (57–81) | 0.522 |
| **Clot burden score, n (%)** | | | |
| CBS 0–9 | 39 (42) | 61 (59) | 0.023 |
| CBS 10 | 53 (58) | 43 (41) | |
| NIHSS, median (IQR) | 10 (7–13) | 7 (4–13) | 0.042 |
| **Medication at enrollment, n (%)** | | | |
| Antihypertensive therapy | 69 (75) | 66 (63) | 0.082 |
| Antiplatelet drug | 36 (39) | 41 (45) | 0.967 |
| Anticoagulant drug | 4 (4) | 7 (7) | 0.469 |
| Lipid lowering therapy | 25 (27) | 22 (21) | 0.325 |
| Antidiabetic therapy | 10 (11) | 18 (17) | 0.199 |
| **Laboratory measurements, median (IQR)** | | | |
| INR | 0.98 (0.94–1.05) | 0.98 (0.93–1.02) | 0.242 |
| APTT (sec) | 27.2 (25.2–31.1) | 27.9 (26.2–30.8) | 0.464 |
| WBC (G/L) | 7.7 (6.3–8.7) | 8.0 (6.5–9.9) | 0.153 |
| RBC (T/L) | 4.5 (4.1–4.8) | 4.6 (4.2–4.9) | 0. 268 |
| Hemoglobin (g/L) | 136.0 (124.0–146.5) | 139.0 (124.0–152.0) | 0.114 |
| Platelets (G/L) | 214 (175–257) | 215 (180–265) | 0.967 |
| Serum glucose (mmol/l) | 6.4 (5.7–7.5) | 6.6 (5.7–8.3) | 0.251 |
| hsCRP (mg/L) | 3.0 (1.8–6.9) | 3.9 (1.7–7.6) | 0.675 |
| Creatinine (umol/L) | 78.5 (67.0–91.0) | 77.5 (67.0–91.0) | 0.938 |
| Plasminogen activity on admission (% | 99.5 (85.3–122.3) | 107.0 (93.0–132.0) | 0.031 |
| Plasminogen activity at 24 h (%) | 89.5 (76.3–104.5) | 90.0 (80.0–105.3) | 0.302 |
| α2-plasmin inhibitor activity on admission (%) | 99.0 (89.5–108.5) | 101.00 (93.3–110.8) | 0.347 |
| α2-plasmin inhibitor activity at 24 h (%) | 75.0 (67.3–87.8) | 76.5 (66.0–85.3) | 0.737 |
| D-dimer level on admission (mg/L) | 0.7 (0.5–1.3) | 0.7 (0.5–1.4) | 0.841 |
| D-dimer level at 24 h (mg/L) | 2.3 (1.1–3.8) | 1.8 (1.1–3.9) | 0.879 |
| Fibrinogen level on admission (g/L) | 3.9 (3.3–4.5) | 3.9 (3.4–4.6) | 0.668 |
| Fibrinogen at 24 h (g/L) | 3.7 (2.9–4.4) | 3.8 (3.2–4.3) | 0.642 |
| Factor XIII activity on admission (%) mean (SD) | 121.4±34.9 | 126.1±37.0 | 0.371 |

(*Continued*)

**Table 2.** (Continued)

|  | Favorable outcome | Unfavorable outcome | p |
|---|---|---|---|
| **Factor XIII activity at 24 h (%)** | 106.2±31.6 | 108.7±39.3 | 0.624 |
| **Stroke etiology (TOAST), n (%)** | | | |
| Large-artery atherosclerosis | 7 (8) | 11 (11) | 0.326 |
| Small-vessel occlusion | 40 (43) | 53 (51) | |
| Cardioembolic | 14 (15) | 17 (16) | |
| Other/undetermined | 31 (34) | 23 (22) | |
| **Imaging data, n (%)** | | | |
| **ASPECTS on admission** | | | |
| 0–7 | 0 (0) | 4 (4) | 0.123 |
| 8–10 | 92 (100) | 97 (93) | |
| **ASPECTS at 24h after thrombolysis** | | | |
| 0–7 | 13 (14) | 45 (43) | <0.001 |
| 8–10 | 79 (86) | 55 (53) | |
| **Factor XIII-A Leu34 carrier n (%)** | 47 (51) | 41 (39) | 0.101 |

APTT, activated partial thromboplastin time; aSICH, asymptomatic intracerebral hemorrhage; ASPECTS, The Alberta Stroke Program early CT score; CBS, clot burden score; ECASS II, European Co-operative Acute Stroke Study-II; HGB, hemoglobin; hsCRP high sensitive CRP; INR, international normalized ratio; IQR, interquartile range, mRS, modified Rankin Scale; NIHSS, National Institutes of Health Stroke Scale; RBC, red blood cell; rt-PA, recombinant tissue plasminogen activator; SD, standard deviation; SICH, symptomatic intracerebral hemorrhage; TOAST, Trial of ORG 10172 in Acute Stroke Treatment; WBC white blood cell

admission or 24 h later. In a previous study by our group involving 131 AIS patients, we found that PAI-activity and antigen levels before thrombolysis, 1 hour and 24 hours after thrombolysis showed no association with short-term or long-term functional outcomes [29], thus levels of this marker were not measured in this cohort. In this patient cohort, plasminogen levels were significantly lower in case of higher CBS (smaller thrombi) that might be a result of significantly higher incorporation of plasminogen to thrombi and as a consequence, the consumption of the protein. Lower plasminogen levels were associated not only with smaller clot burden but with better short-term functional outcomes in an univariate model. On the other hand, in a multivariate model, the independent effect of admission plasminogen levels on short-term outcomes was not proved. In the literature only one study can be found where the relation of hemostasis proteins and CBS was assessed [30]. In this report only fibrinogen was measured in a smaller subset of AIS patients, and higher fibrinogen levels were found to be associated with smaller clot burden. In our study of much larger sample size, admission and 24 h post-lysis fibrinogen levels were practically identical in patient groups of different CBS,

**Table 3. Independent predictors of poor short-term outcome[a] in the studied cohort.**

|  | OR | 95%CI | p |
|---|---|---|---|
| **Age** | 1.026 | 1.000 to 1.053 | 0.049 |
| **NIHSS on admission** | 0.916 | 0.861 to 0.974 | 0.005 |
| **Clot burden score (0–9 vs. 10)** | 2.777 | 1.439 to 5.361 | 0.002 |

Last step of backward multiple regression analysis is provided.

[a]Poor short-term outcome is defined as a less than 4 points decrease or any increase of NIHSS or death by day 7 post-event. Backward multiple regression model included age, sex, previous stroke, NIHSS on admission, plasminogen activity on admission, clot burden score (CBS 0–9 vs CBS 10). 95%CI, 95% confidence interval; NIHSS, National Institutes of Health Stroke Scale; OR, odds ratio.

**Table 4. Baseline parameters and specific hemostasis markers according to long-term functional outcomes in the studied cohort.**

| | mRS 0–2 | mRS 3–6 | p |
|---|---|---|---|
| **Number of patients** | 101 | 92 | |
| **Age (years), median (IQR)** | 67 (60–75) | 73 (66–80) | <0.01 |
| **Male, n (%)** | 56 (55) | 46 (50) | 0.449 |
| **Cerebrovascular risk factors, n (%)** | | | |
| Arterial hypertension | 85 (84) | 81 (88) | 0.437 |
| Atrial fibrillation | 19 (19) | 21 (23) | 0.492 |
| Previous stroke | 22 (22) | 26 (28) | 0.281 |
| Hyperlipidaemia | 62 (61) | 55 (60) | 0.820 |
| Diabetes mellitus | 17 (17) | 33 (36) | 0.003 |
| **Smoking, n (%)** | | | |
| Non-smoker | 62 (61) | 64 (70) | 0.416 |
| Previous smoker | 8 (8) | 9 (10) | |
| Current smoker | 26 (26) | 17 (18) | |
| Undetermined | 5 (5) | 2 (2) | |
| **BMI, median (IQR)** | 27.2 (24.3–30.5) | 27.7 (24.3–32.1) | 0.374 |
| **Duration of thrombolysis (min), median (IQR)** | 60 (60–60) | 60 (60–60) | 0.770 |
| **Time-to-treatment (min), mean (SD)** | 143 (102–180) | 145 (116–180) | 0.488 |
| **rt-PA dose (mg), median (IQR)** | 69 (57–80) | 68 (57–82) | 0.847 |
| **Clot burden score, n (%)** | | | |
| CBS 0–9 | 34 (33) | 61 (66) | <0.001 |
| CBS 10 | 67 (67) | 31 (34) | |
| **NIHSS, median (IQR)** | 7 (4–10) | 11.5 (8–15) | <0.001 |
| **Medication at enrollment, n (%)** | | | |
| Antihypertensive therapy | 64 (64) | 70 (76) | 0.055 |
| Antiplatelet drug | 34 (34) | 41 (45) | 0.121 |
| Anticoagulant drug | 7 (7) | 4 (4) | 0.440 |
| Lipid lowering therapy | 20 (220) | 27 (29) | 0.123 |
| Antidiabetic therapy | 11 (11) | 18 (20) | 0.092 |
| **Laboratory measurements, median (IQR)** | | | |
| INR | 0.97 (0.93–1.02) | 0.99 (0.93–1.04) | 0.369 |
| APTT (sec) | 27.7 (25.5–31.1) | 27.7 (25.9–30.5) | 0.930 |
| WBC (G/L) | 8.1 (6.5–9.0) | 7.8 (6.4–10.0) | 0.962 |
| RBC (T/L) | 4.6 (4.2–4.9) | 4.5 (4.1–4.9) | 0.197 |
| Hemoglobin (g/L), | 138.5 (127.0–150.0) | 135.0 (120.0–145.0) | 0.066 |
| Platelets (G/L) | 210 (178–255) | 217 (177–269) | 0.490 |
| Serum glucose (mmol/l) | 6.3 (5.4–7.2) | 6.8 (5.9–8.6) | 0.003 |
| hsCRP (mg/L) | 2.8 (1.7–5.4) | 4.8 (1.8–11.7) | 0.016 |
| Creatinine (umol/L) | 75.0 (66.0–86.5) | 85.5 (69.3–100.0) | 0.011 |
| Plasminogen activity on admission (% | 102.5 (89.3–127.0) | 107.0 (90.1–127.8) | 0.586 |
| Plasminogen activity at 24 h (%) | 90.0 (81.5–107.5) | 89.0 (76.8–102.3) | 0.107 |
| α2-plasmin inhibitor activity on admission (%) | 101.0 (90.5–109.0) | 100.5 (92.5–110.3) | 0.850 |
| α2-plasmin inhibitor activity at 24 h (%) | 76.0 (68.5–86.5) | 73.5 (65.8–83.3) | 0.238 |
| D-dimer level on admission (mg/L) | 0.66 (0.49–0.99) | 0.85 (0.53–1.63) | 0.012 |
| D-dimer level at 24 h (mg/L) | 1.74 (1.1–3.1) | 2.5 (1.2–4.6) | 0.060 |
| Fibrinogen level on admission (g/L) | 3.7 (3.3–4.3) | 4.1 (3.5–4.8) | 0.020 |
| Fibrinogen at 24 h (g/L) | 3.8 (3.2–4.4) | 3.7 (2.94.3) | 0.374 |
| Factor XIII activity on admission (%) mean (SD) | 124.7±36.1 | 121.4±35.7 | 0.542 |

*(Continued)*

**Table 4.** (Continued)

| | mRS 0–2 | mRS 3–6 | p |
|---|---|---|---|
| Factor XIII activity at 24 h (%) | 117.2 (88.9–132.6) | 99.1 (81.5–122.4) | 0.025 |
| **Stroke etiology (TOAST), n (%)** | | | |
| Large-artery atherosclerosis | 12 (11) | 4 (4) | 0.035 |
| Small-vessel occlusion | 38 (38) | 52 (57) | |
| Cardioembolic | 17 (17) | 14 (15) | |
| Other/undetermined | 34 (34) | 22 (24) | |
| **Imaging data, n (%)** | | | |
| **ASPECTS on admission** | | | |
| 0–7 | 0 (0) | 3 (3) | 0.101 |
| 8–10 | 100 (99) | 85 (92) | |
| **ASPECTS at 24h after thrombolysis** | | | |
| 0–7 | 14 (13) | 42 (46) | <0.001 |
| 8–10 | 87 (87) | 44 (48) | |
| Factor XIII-A Leu34 carrier n (%) | 42 (42) | 43 (47) | 0.471 |

APTT, activated partial thromboplastin time; aSICH, asymptomatic intracerebral hemorrhage; ASPECTS, The Alberta Stroke Program early CT score; CBS, clot burden score; ECASS II, European Co-operative Acute Stroke Study-II; HGB, hemoglobin; hsCRP high sensitive CRP; INR, international normalized ratio; IQR, interquartile range, mRS, modified Rankin Scale; NIHSS, National Institutes of Health Stroke Scale; RBC, red blood cell; rt-PA, recombinant tissue plasminogen activator; SD, standard deviation; SICH, symptomatic intracerebral hemorrhage; TOAST, Trial of ORG 10172 in Acute Stroke Treatment; WBC white blood cell

therefore, this association could not be confirmed. Apart from the fibrin and fibrinolytic network, thrombus size might be influenced by red blood cells and platelets that are important components of thrombi. Here we show that slightly, but significantly decreased red blood cell count and hemoglobin concentration was found in patients with higher thrombus burden (CBS<10). Without further experiments, it is difficult to interpret whether this difference might be the result of consumption, but overall, red blood cell count was not associated with outcomes in this cohort. Platelet count did not show an association with thrombus burden or outcomes.

Here we show for the first time that FXIII-A Val34Leu polymorphism has a significant effect on clot burden in vivo. FXIII has been implicated in limiting thrombus size by various mechanisms, including the regulation of the retention of red blood cells in thrombi or by

**Table 5. Independent predictors of unfavorable long-term outcome[a] (mRS) in the total cohort.**

| | OR | 95%CI | p |
|---|---|---|---|
| **Creatinine level on admission** | 1.019 | 1.001 to 1.038 | 0.043 |
| **NIHSS on admission** | 1.119 | 1.039 to 1.205 | 0.003 |
| **Age** | 1.033 | 1.000 to 1.066 | 0.048 |
| **Clot burden score (0–9 vs. 10)** | 2.501 | 1.179 to 5.306 | 0.017 |

Last step of backward multiple regression analysis is provided.

[a]Unfavorable long-term outcome is defined as mRS 3–6 by day 90 post-event. Backward multiple regression model included age, sex, diabetes mellitus, NIHSS on admission, clot burden score (CBS 0–9 vs CBS 10), admission glucose level, hsCRP, creatinine level on admission, D-dimer level on admission, fibrinogen level on admission, FXIII activity at 24 hours post-event, TOAST values. 95%CI, 95% confidence interval; hsCRP high sensitive CRP, mRS, modified Rankin Scale; NIHSS, National Institutes of Health Stroke Scale; OR, odds ratio, TOAST, Trial of ORG 10172 in Acute Stroke Treatment.

down-regulating the accumulation of platelets on fibrin, among others [16, 17]. Of the major FXIII polymorphisms, the FXIII-A Val34Leu polymorphism has been investigated most extensively in relation with the risk of thrombotic diseases. It has been confirmed by meta-analyses that the Leu34 allele provides a moderate protection against coronary artery disease and venous thromboembolism, but evidence is still lacking in case of ischemic stroke [16]. The effect of FXIII-A Val34Leu polymorphism has been known to be influenced by complex gene-environmental interactions [31]. A number of reports have shown that the protective effect of the Leu34 prevails at high fibrinogen concentrations [27, 32]. Experimental reports demonstrated that in the presence of the Leu34 allele the formed fibrin is more susceptible to fibrinolysis than fibrin formed in the presence of Val34 [16]. Still, explaining the biochemical link between earlier FXIII activation provided by the Leu34 allele and protection against thrombotic events remains a challenge. In a most recent report, we have shown for the first time that in in vitro experiments, reconstituted whole blood clot mass is greatly determined by the FXIII-A Val34 Leu polymorphism [17]. Using plasma samples with high fibrinogen levels (>3.5 g/L), clot mass was higher for clots with Val34 as compared with clots with homozygous Leu34. In the current cohort of AIS patients, median fibrinogen level on admission was above 3.5 g/L in both groups of patients (3.88 g/L vs. 3.91 g/L in CBS 0–9 vs. CBS 10), therefore the effect of Leu34 allele could potentially prevail. Further statistical analysis of the FXIII-A Leu34 allele's effect on clot burden with respect to fibrinogen levels did not reveal statistically significant differences among groups, which might be explained by the influence of the acute event on fibrinogen levels. Here we show that the presence of the Leu34 allele provided a significant protective effect against developing larger (CBS 0–9) clots (OR: 0.519; 95%CI: 0.298–0.922, p = 0.0227). Interestingly, in multivariate analysis, the polymorphism did not prove to be an independent predictor of functional outcomes. As thrombolysis outcomes are influenced by several factors, most importantly by the severity and localization of stroke, it is plausible that the FXIII-A Val34Leu polymorphism is not a significant contributor of the overall functional outcome of patients. This is in line with our previous report where FXIII-A Val34Leu had no effect on outcomes in a smaller AIS patient cohort undergoing thrombolysis treatment [33]. In that report, other major FXIII polymorphisms, FXIII-A p.Tyr204Phe (c.614 A > T; rs3024477), FXIII-B p.His95Arg (c.344 G > A; rs6003) and FXIII-B Intron K (IVS11 c.1952 + 144 C > G; rs12134960) were also investigated and they showed no association with stroke severity, unfavorable outcomes of therapy, therapy-associated symptomatic intracranial hemorrhage and mortality.

Based on multivariate models, CBS was found to be a significant independent predictor of short-term and long-term outcomes in the studied cohort of AIS patients treated by intravenous thrombolysis (CBS 0–9 vs. 10: OR: 2.777; 95%CI: 1.439–5.361, p = 0.002 and OR: 2.501; 95%CI: 1.179–5.306, p = 0.017, respectively). It has been proposed that patients with lower CBS (larger thrombi) are more likely to benefit from endovascular treatment as opposed to intravenous thrombolysis alone [34, 35]. On the other hand, several lines of evidence suggest that besides thrombus length, clot characteristics also play an important role in the likelihood of recanalization with intravenous thrombolysis and/or thrombectomy [36, 37]. Here we show that thrombus size is indeed among the most important factors for the outcome of intravenous thrombolysis, and potential differences between cellular composition or the levels of key fibrinolysis factors in larger or smaller thrombi cannot be detected using peripheral blood samples of patients on admission and 24 h post-lysis. In univariate models, the association of higher admission D-dimer levels, higher admission fibrinogen levels and lower FXIII activity 24 h post-lysis were significantly associated with poor-long term outcomes. However, in a multiple logistic regression model including CBS, the association of these hemostasis proteins or fibrinolysis markers with long-term outcomes could not be revealed.

### Limitations of the study

Results of the present study should be interpreted in the context of its limitations and strengths. Due to the single-centered study design, the sample size is limited, however, as compared to other published studies measuring hemostasis or fibrinolysis biomarkers in acute ischemic stroke patients from pre-thrombolysis and post-thrombolysis samples, it is among the largest studies as yet [38]. Being single-centered, our study had the advantages of uniform sample handling and uniform patient care, moreover, the proportion of patients that were lost to follow-up was remarkably low (2% and 3.5% for short-term and long-term follow up, respectively) as compared to that observed in other studies involving post-stroke patients [38]. It must be noted, however, that the relatively low sample size together with this drop-out rate at follow-up might have influenced the results to a certain extent and thus larger clinical studies are needed to confirm and to validate our data.

As the in vitro described whole blood clot mass reducing effects of the FXIII-A Leu34 allele has not been proven in conditions resembling acute thromboembolism, it must be acknowledged as a limitation that conclusions of this study based on previously reported in vitro data may be speculative to a certain extent.

## Conclusions

To conclude, CBS was found to be a significant independent predictor of short-term and long-term outcomes in the studied cohort of AIS patients treated by intravenous thrombolysis. FXIII-A 34Leu carrier status was found to be associated with smaller thrombus burden, which is consistent with the *in vitro* described whole blood clot mass reducing effects of the allele, but the polymorphism had no effect on thrombolysis outcomes.

## Supporting information

**S1 Table. Effects of fibrinogen and FXIII-A Val34 or Leu34 alleles on clot burden score.** (DOCX)

**S2 Table. Levels of different hemostasis markers according to CBS, median (IQR).** (DOCX)

## Author Contributions

**Conceptualization:** László Csiba, Zsuzsa Bagoly.

**Data curation:** István Szegedi, Rita Orbán-Kálmándi, Ferenc Sarkady, Nikolett Vasas, Ervin Berényi, Alexandra Crişan, Zsuzsa Bagoly.

**Formal analysis:** István Szegedi, Rita Orbán-Kálmándi, Attila Nagy, Nikolett Vasas, Máté Sik, Levente István Lánczi, Ervin Berényi, Alexandra Crişan, Zsuzsa Bagoly.

**Funding acquisition:** László Csiba, Zsuzsa Bagoly.

**Investigation:** István Szegedi, Rita Orbán-Kálmándi, Ferenc Sarkady, Zsuzsa Bagoly.

**Supervision:** László Csiba, Zsuzsa Bagoly.

**Writing – original draft:** István Szegedi, Zsuzsa Bagoly.

**Writing – review & editing:** László Oláh, László Csiba, Zsuzsa Bagoly.

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
