## [Decision Letter · Decision Letter 0]

6 May 2021

PONE-D-21-10757

Decreased clot burden is associated with factor XIII Val34Leu polymorphism and better functional outcomes in acute ischemic stroke patients treated with intravenous thrombolysis

PLOS ONE

Dear Dr. Bagoly,

Thank you for submitting your manuscript to PLOS ONE. After careful consideration, we feel that it has merit but does not fully meet PLOS ONE’s publication criteria as it currently stands. Therefore, we invite you to submit a revised version of the manuscript that addresses the points raised during the review process.

We look forward to receiving your revised manuscript.

Kind regards,

Arijit Biswas

Academic Editor

PLOS ONE

Additional Editor Comments:

The manuscript by Szegedi et al. aims at assessing the relevance of clot burden score and other parameters on the outcome of thrombolysis in acute ischemic stroke patient. Two reviewers have critically analyzed the manuscript and proposed some changes. Reviewer 1 has suggested minor revision while Reviewer 2 suggests major revisions. I agree with the suggestions made by both reviewers. However, none of the reviewers have suggested any major new experimentation to be performed and the changes suggested are more in the form of formatting or extending the discussion in certain context (like the context of Val34Leu polymorphism). Similarly certain questions raised by the reviewers can also be answered by the author as a point-wise reply or with specific edits in the manuscript and does not require wholesome changes to the manuscript itself. In my view therefore, the proposed changes fall in the purview of minor revisions only, although ofcourse the authors need to make these changes and answer the questions absolutely. From my personal perspective I find the article of novel relevance and quite well written and well reported. Apart from the questions raised by the two reviewers, I have a small query (more like a comment) which is that the authors have focussed significantly on the val34leu polymorphism and ofcourse for good reason. Could the authors briefly mention the relevance (or the lack of it) of other background genetic polymorphisms of FXIII (if any) in the context of thrombolytic outcomes in acute ischemic stroke? If there are no studies in this regard, there may be a suggestion for investigating them in this context unless they are simply in strong linkage disequilibrium with this (or any other functional polymorphism).

Journal Requirements:

Reviewers' comments:

Reviewer's Responses to Questions

**Comments to the Author**

1. Is the manuscript technically sound, and do the data support the conclusions?

Reviewer #1: Yes

Reviewer #2: Yes

2. Has the statistical analysis been performed appropriately and rigorously? 

Reviewer #1: Yes

Reviewer #2: Yes

3. Have the authors made all data underlying the findings in their manuscript fully available?

Reviewer #1: Yes

Reviewer #2: Yes

4. Is the manuscript presented in an intelligible fashion and written in standard English?

Reviewer #1: Yes

Reviewer #2: Yes

5. Review Comments to the Author

Reviewer #1: This manuscript by Szegedi et al. aims at assessing the relevance of clot burden score and other parameters on the outcome of thrombolysis in acute ischemic stroke patient.

This manuscript is very nicely written, with well-presented and analysed data.

The reviewers only has one main comment: Would the author be able to speculate on which clot burden score would be a reasonable threshold in determining the appropriate treatment (thrombolysis or thrombectomy) in AIS patients.

Minor comment: in your results section, you only have one subtitle “Baseline characteristics of patients according to CBS”. You should either take this out, or add other subsections.

Reviewer #2: The present study demonstrates for the first time that in acute ischemic stroke patients with a common genetic polymorphism, FXIII-A Leu34, smaller thrombus burden is observed. The study has novel aspects with regard to search for factors affecting thrombus size and thrombolysis outcomes in acute stroke. The laboratory methods used were standard. Clinical evaluation of the patients enrolled was sufficient.

My concerns are as follows:

1. The frequency of the 3 allelic variants should be presented. Were they in Hardy Weinberg equilibirium?

2. The Lancet study of Ariens' group in 2003 demonstrated a strong impact of plasma fibrinogen concentration on the FXIII-A Leu34 allele on fibrin properties. Did the authors assess their fidings with respect to fibrinogen?

3. The authors even in the conclusions of the abstract addressed "the in vitro described whole blood clot mass reducing effects" of the Leu34 allele. To the knowledge of this reviewer, the vitro data mentioned have not been performed in acute thromboembolism where activation of blood coagulation and enhanced inflammatory state along with oxidative stress occur. Since in the current study no investigations to support the above statement in acute stroke were presented, that conclusion is overly speculative. This is a major limitation which should be ackowledged; in fact the section of Study limiatations is missing in the masnuscript and should be added including comment on the risk of underpowered analysis in allele-associated analysis).

4. The study shows that the FXIII-A Leu34 polymorphism has no impact on thrombolysis outcomes in acute stroke despite association with thrombus burden. This aspect should be discussed in more detail.

5.Table 2 did not show relevant intergroup differences and a brief comment would be sufficient. The same holds true for Table 4. Tables 1 and 3 could be combined since most variables presented were identical in both.

6. Did the authors follow the patients for a longer period of time? Were any data on recurrent stroke available?

Minor comments

In table 1 BMI should be replaced by obesity; BMI is not a risk factor for stroke, only its increased values.

In the first paragraph of the discussion, the authors stated that "thrombus size directly

relates to major coagulation or fibrinolysis proteins regulating clot structure and lysis". Obviously, apart from fibrin network, red blood cells and platelets are important components of each thrombus. Hemoglobin and erythrocytes should be added to tables.

The authors claimed that they showed "major fibrinolysis parameters". PAI_1 is of key importance, but it has not been determined, therefore more precision is suggested while discussing fibrinolysis.

Reference style requires correction.

6. PLOS authors have the option to publish the peer review history of their article (what does this mean?). If published, this will include your full peer review and any attached files.

Reviewer #1: No

Reviewer #2: No

---

## [Author Response · Author response to Decision Letter 0]

18 Jun 2021

Response to Reviewers 

We are grateful to the Reviewers for the critical reading and overall positive assessment of our manuscript and the valuable remarks.

 We addressed all comments raised and we believe that the changes introduced in the manuscript resulted in an improved version of the paper. 

Response to the Editor

„Apart from the questions raised by the two reviewers, I have a small query (more like a comment) which is that the authors have focussed significantly on the val34leu polymorphism and ofcourse for good reason. Could the authors briefly mention the relevance (or the lack of it) of other background genetic polymorphisms of FXIII (if any) in the context of thrombolytic outcomes in acute ischemic stroke? If there are no studies in this regard, there may be a suggestion for investigating them in this context unless they are simply in strong linkage disequilibrium with this (or any other functional polymorphism).”

We are thankful to the Editor for the critical reading of our manuscript and for the valuable remark. The relevance of other major FXIII polymorphisms was investigated in detail in another paper by our group (Szekely et al, Sci Rep 2018; 8:7662). In that report including 132 consecutive AIS patients receiving i.v thrombolysis, other major FXIII polymorphisms, including FXIII-A p.Tyr204Phe (c.614 A > T; rs3024477), FXIII-B p.His95Arg (c.344 G > A; rs6003) and FXIII-B Intron K (IVS11 c.1952 + 144 C > G; rs12134960) were also investigated and they showed no association with stroke severity, unfavorable outcomes of therapy, therapy-associated symptomatic intracranial hemorrhage and mortality in that cohort.

This is now stated in the Discussion section of the manuscript, page 18, lines 22-28.

Szekely EG, Czuriga-Kovacs KR, Bereczky Z, Katona E, Mezei ZA, Nagy A, et al. Low factor XIII levels after intravenous thrombolysis predict short-term mortality in ischemic stroke patients. Sci Rep. 2018;8(1):7662. doi: 10.1038/s41598-018-26025-z. PubMed PMID: 29769590; PubMed Central PMCID: PMCPMC5955963.

Response to Reviewer 1

We would like to thank the Reviewer the overall positive comments of our manuscript. 

Response to the main comment:

„Would the author be able to speculate on which clot burden score would be a reasonable threshold in determining the appropriate treatment (thrombolysis or thrombectomy) in AIS patients.”

This is a very interesting question indeed. As the study was not planned to answer this specific question, the wording „speculate” is very appropriate here and unfortunately we cannot provide a valid statistical analysis to answer the question. We performed an analysis where the studied population was layered based on their clot burden score (e.g. CBS 10 vs. 9-0; 10-9 vs. 8-0; 10-8 vs. 7-0; etc.) and the score where the benefit of thrombolysis was not statistically significant anymore was at CBS 2. This would mean that the only cases where thrombolysis would not have a benefit is CBS ≤2, although it must be emphasized that bleeding complications were not incorporated in this analysis for the lack of statistical power. Based on the 2018 Guideline of AHA/ASA guideline for the early management of patients with acute ischemic stroke, it is an IA level of evidence and recommendation that patients eligible for IV alteplase should receive IV alteplase even if EVTs are being considered [1]. Therefore, at least based on current evidence, CBS itself does not alter this decision. On the other hand, a low CBS could indicate the high probability of unsuccessful thrombolysis to the clinicians and suggests the need to organize and perform mechanical thrombectomy after thrombolysis.

1. Powers WJ, Rabinstein AA, Ackerson T, Adeoye OM, Bambakidis NC, Becker K, et al. 2018 Guidelines for the Early Management of Patients With Acute Ischemic Stroke: A Guideline for Healthcare Professionals From the American Heart Association/American Stroke Association. Stroke. 2018;49(3):e46-e110. Epub 2018/01/26. doi: 10.1161/STR.0000000000000158. PubMed PMID: 29367334.

Minor comment: 

„ in your results section, you only have one subtitle “Baseline characteristics of patients according to CBS”. You should either take this out, or add other subsections. „

Thank you for noticing this, we deleted the subtitle of the subsection.

Response to Reviewer 2

We would like to thank the Reviewer the overall positive comments of our manuscript. 

Major comments:

 The frequency of the 3 allelic variants should be presented. Were they in Hardy Weinberg equilibirium?

Genotype frequencies of FXIII-A Val34Leu polymorphism were consistent with Hardy-Weinberg equilibrium in the total cohort (FXIII-A Val34Val: n=112 (56%), FXIII-A Val34Leu: n=78 (39%) and FXIII-A Leu34Leu: n= 10 (5%). This information is now included in the Results section of the manuscript, page 6, ultimate sentence.

2. The Lancet study of Ariens' group in 2003 demonstrated a strong impact of plasma fibrinogen concentration on the FXIII-A Leu34 allele on fibrin properties. Did the authors assess their fidings with respect to fibrinogen?

In order to better reply to this comment, we performed a statistical analysis in which the specific relationships between admission fibrinogen concentration (below or above 3.5 g/L) and the CBS for each FXIII-A genotype was studied, but no significant difference was found between groups. Results are described on page 7, lines 6-10 and Table S1 is added to the manuscript.

Given the modifying effects of fibrinogen concentration on FXIII-A Val34Leu genotype dependent clot structure and thrombus burden, we further looked at the specific relationships between admission fibrinogen concentration (below or above 3.5 g/L) and the CBS for each FXIII-A genotype, but no significant difference was found between groups (Table S2).

Also, the Discussion section is now supplemented with the following reasoning (page 18, lines 10-17):

In the current cohort of AIS patients, median fibrinogen level on admission was above 3.5 g/L in both groups of patients (3.88 g/L vs. 3.91 g/L in CBS 0-9 vs. CBS 10), therefore the effect of Leu34 allele could potentially prevail. Further statistical analysis of the FXIII-A Leu34 allele’s effect on clot burden with respect to fibrinogen levels did not reveal statistically significant differences among groups, which might be explained by the influence of the acute event on fibrinogen levels.

We also included the mentioned reference as a citation. 

Lim BC, Ariens RA, Carter AM, Weisel JW, Grant PJ. Genetic regulation of fibrin structure and function: complex gene-environment interactions may modulate vascular risk. Lancet. 2003;361(9367):1424-31. Epub 2003/05/03. doi: 10.1016/S0140-6736(03)13135-2. PubMed PMID: 12727396.

3. The authors even in the conclusions of the abstract addressed "the in vitro described whole blood clot mass reducing effects" of the Leu34 allele. To the knowledge of this reviewer, the vitro data mentioned have not been performed in acute thromboembolism where activation of blood coagulation and enhanced inflammatory state along with oxidative stress occur. Since in the current study no investigations to support the above statement in acute stroke were presented, that conclusion is overly speculative. This is a major limitation which should be ackowledged; in fact the section of Study limiatations is missing in the masnuscript and should be added including comment on the risk of underpowered analysis in allele-associated analysis).

This is very important point, indeed. A Limitation section is now included in the manuscript, acknowledging the above mentioned limitations of the study (page 19):

Limitations of the study

Results of the present study should be interpreted in the context of its limitations and strengths. Due to the single-centered study design, the sample size is limited, however, as compared to other published studies measuring hemostasis or fibrinolysis biomarkers in acute ischemic stroke patients from pre-thrombolysis and post-thrombolysis samples, it is among the largest studies as yet. [1] Being single-centered, our study had the advantages of uniform sample handling and uniform patient care, moreover, the proportion of patients that were lost to follow-up was remarkably low (2% and 3.5% for short-term and long-term follow up, respectively) as compared to that observed in other studies involving post-stroke patients [1]. It must be noted, however, that the relatively low sample size together with this drop-out rate at follow-up might have influenced the results to a certain extent and thus larger clinical studies are needed to confirm and to validate our data.

As the in vitro described whole blood clot mass reducing effects of the FXIII-A Leu34 allele has not been proven in conditions resembling acute thromboembolism, it must be acknowledged as a limitation that conclusions of this study based on previously reported in vitro data may be speculative to a certain extent. 

1. Donkel SJ, Benaddi B, Dippel DWJ, Ten Cate H, de Maat MPM. Prognostic Hemostasis Biomarkers in Acute Ischemic Stroke. Arterioscler Thromb Vasc Biol. 2019;39(3):360-72. Epub 2019/02/01. doi: 10.1161/ATVBAHA.118.312102. PubMed PMID: 30700129; PubMed Central PMCID: PMCPMC6392207.

4. The study shows that the FXIII-A Leu34 polymorphism has no impact on thrombolysis outcomes in acute stroke despite association with thrombus burden. This aspect should be discussed in more detail.

We would like to thank the Reviewer for this useful comment. This aspects is now discussed in more detail in the Discussion section: page 18, lines 18-23):

As thrombolysis outcomes are influenced by several factors, most importantly by the severity and localization of stroke, it is plausible that the FXIII-A Val34Leu polymorphism is not a significant contributor of the overall functional outcome of patients. This is in line with our previous report where FXIII-A Val34Leu had no effect on outcomes in a smaller AIS patient cohort undergoing thrombolysis treatment [3]. 

5. Table 2 did not show relevant intergroup differences and a brief comment would be sufficient. The same holds true for Table 4. Tables 1 and 3 could be combined since most variables presented were identical in both.

Table 2 was deleted from the manuscript as requested and added as a Supplemental Table. We kept Table 4 in the manuscript as we believe that it indeed has relevant, significant intergroup differences.

Unfortunately, Tables 1 and 3 cannot be combined as the colums of these Tables represent completely different groups (i.e. Table 1: CBS 0-9 vs.CBS 10 while Table 3 (Table 2 in the revised version of the manuscipt): favorable vs. unfavorable short term outcomes). 

6. Did the authors follow the patients for a longer period of time? Were any data on recurrent stroke available?

Unfortunately, we did not have the opportunity to follow the patients for a longer period of time. However, in most clinical studies on acute ischemic stroke, it is a standard practice to follow the patients for 90 days and to determine the long-term functional outcome of the acute event by the mRS at this point. Beyond this time, outcomes might be determined by several factors unrelated to the acute event itself (including co-morbidities, treatment of risk factors, socio-economic factors, etc), which also potentially influence recurrent stroke, therefore this aspect was not investigated and was out of the scope of this study.

Minor comments:

„In table 1 BMI should be replaced by obesity; BMI is not a risk factor for stroke, only its increased values.”

Thank you for noticing this error, BMI was removed in every Table from the section „Cerebrovascular risk factors” and was placed below as an individual parameter. 

„In the first paragraph of the discussion, the authors stated that "thrombus size directly

relates to major coagulation or fibrinolysis proteins regulating clot structure and lysis". Obviously, apart from fibrin network, red blood cells and platelets are important components of each thrombus. Hemoglobin and erythrocytes should be added to tables.” 

Hemoglobin and red blood cell count were added to the Tables 1,2, and 4. Slightly, but significantly decreased red blood cell count and hemoglobin concentration was found in patients with higher thrombus burden. Results were discussed at the end of the first paragraph of the Discussion (page 17, lines 20-27): 

Apart from the fibrin and fibrinolytic network, thrombus size might be influenced by red blood cells and platelets that are important components of thrombi. Here we show that slightly, but significantly decreased red blood cell count and hemoglobin concentration was found in patients with higher thrombus burden (CBS<10). Without further experiments, it is difficult to interpret whether this difference might be the result of consumption, but overall, red blood cell count was not associated with outcomes in this cohort. Platelet count did not show an association with thrombus burden or outcomes.

„The authors claimed that they showed "major fibrinolysis parameters". PAI_1 is of key importance, but it has not been determined, therefore more precision is suggested while discussing fibrinolysis.

Reference style requires correction.”

We would like to thank the reviewer for noticing the fact that we did not include or mention PAI-1 in this paper. In a previous study by our group involving 131 AIS patients, we found that PAI-activity and antigen levels before thrombolysis, 1 hour and 24 hours after thrombolysis showed no association with short‐term or long‐term functional outcomes (Szegedi et al, Ann Clin Transl Neurol, 2019;6(11):2240-50), thus levels of this marker were not measured in this cohort. We are thankful to the reviewer for noticing that we haven’t explained this circumstance in the manuscript and now this exact reasoning is mentioned on page 17, lines 6-9.

„Reference style requires correction.”

In case of all references, EndNote Style for Plos was used when citing relevant papers.

Once again, we are grateful to the Reviewer for the overall positive evaluation of our manuscript and the helpful criticism.

---

## [Editor Report · Decision Letter 1]

24 Jun 2021

Decreased clot burden is associated with factor XIII Val34Leu polymorphism and better functional outcomes in acute ischemic stroke patients treated with intravenous thrombolysis

PONE-D-21-10757R1

Dear Dr. Bagoly,

We’re pleased to inform you that your manuscript has been judged scientifically suitable for publication and will be formally accepted for publication once it meets all outstanding technical requirements.

Kind regards,

Arijit Biswas

Academic Editor

PLOS ONE

---

## [Editor Report · Acceptance letter]

28 Jun 2021

PONE-D-21-10757R1 

Decreased clot burden is associated with factor XIII Val34Leu polymorphism and better functional outcomes in acute ischemic stroke patients treated with intravenous thrombolysis 

Dear Dr. Bagoly:

I'm pleased to inform you that your manuscript has been deemed suitable for publication in PLOS ONE. Congratulations! Your manuscript is now with our production department. 

Kind regards, 

on behalf of

Dr. Arijit Biswas 

Academic Editor

PLOS ONE